# Study on the Effects of Ultrasonic Agitation on CO$_2$ Adsorption Efficiency Improvement of Cement Paste

Lili Liu [1,2], Yongsheng Ji [1,*], Zhanguo Ma [1], Furong Gao [1] and Zhishan Xu [1]

1   Jiangsu Key Laboratory Environmental Impact and Structural Safety in Engineering, China University of Mining and Technology, Xuzhou 221116, China; dxwlll@sina.com (L.L.); zgma@cumt.edu.cn (Z.M.); furonggao@sina.com (F.G.); tb20030006b2@cumt.edu.cn (Z.X.)
2   Xuzhou University of Technology, Xuzhou 221000, China
*   Correspondence: jiyongsheng@cumt.edu.cn; Tel.: +86-516-83995295

**Abstract:** To realize high-efficiency CO$_2$ absorption by fresh cement paste, ultrasonic vibration technology is introduced into the CO$_2$ absorption test device used in this study. Influences of ultrasonic frequency on the CO$_2$ absorption rate (CO$_2$ AR) and the ultimate absorption amount of fresh cement paste are analyzed. Furthermore, the influencing laws of the CO$_2$ absorption amount (CO$_2$ AA) on the fluidity, pore distribution, and mechanical properties of cement paste under ultrasonic vibrating agitation are analyzed by measuring the variations of the CO$_2$ AA of cement paste. Results demonstrate that ultrasonic vibrating agitation not only can increase the CO$_2$ AR and ultimate absorption amount of fresh cement paste, but also can optimize the internal pore structure of materials and compressive strength of cement-based materials.

**Keywords:** ultrasonic vibrating agitation; CO$_2$ absorption rate; CO$_2$ absorption amount; fresh cement paste; mechanical properties





## 1. Introduction

The greenhouse effect which is caused by excessive CO$_2$ emissions has brought many hazards. Global warming has become a primary issue of the top 10 global environmental problems. Nowadays, many methods exist to decrease global CO$_2$ emissions, such as changing the energy structure, chemical absorption, and blocked curing [1]. However, all of these methods have some technological defects. They all incur relatively high costs and cannot solve carbon emission problems within a short period of time. Portland cement is a traditional building material used most extensively around the world which has also attracted wide attention for reducing CO$_2$ emissions [2]. Cement production comes at the cost of great consumption of limestone and fuels, the pyrolysis and combustion of which release CO$_2$. According to the statistics, approximately 1 ton of CO$_2$ emissions during the production of 1 ton of clinkers is produced [3]. From 2013 to 2016, the global cement output exceeded 4 billion tons, equating to about 16 million tons of global CO$_2$ emissions [4,5]. Decreasing CO$_2$ emissions in the Portland cement industry is therefore an important part of the global CO$_2$ emission reduction project [6].

Some studies have pointed out that cement in concrete can produce a significant amount of Ca(OH)$_2$ during the hydration process, accounting for about 20–30% of total hardened cement pastes, and largely exists in the pores of hardened cement pastes in their crystal form. In solution, Ca(OH)$_2$ can extremely easily react with CO$_2$ to produce CaCO$_3$. As a result, concrete has considerable potential for CO$_2$ absorption [7,8]. Compared with mineral carbonisation and ocean storage, increasing the operability and application prospects of CO$_2$ absorption by concrete has recently become a hot research topic. However, realising high-efficiency CO$_2$ absorption by concrete is still a huge challenge [9].

Studies on CO$_2$ absorption of fresh cement pastes have been reported. It is found that the stirring rate, water-cement ratio and addition order of water reducing agents can

significantly influence the $CO_2$ AR and ultimate absorption amount of fresh cement paste significantly. More importantly, the $CO_2$ AR and ultimate absorption amount of fresh cement paste can increase by appropriately setting these parameters [10]. Nevertheless, the stirring time of concrete is very short in practice, which is generally in the range of 30–90 s. Furthermore, reasonable settings of the stirring rate, water-cement ratio and addition order of water reducing agents alone cannot meet the fast $CO_2$ absorption rate required in practical concrete production [11,12]. Hence, the $CO_2$ absorption efficiency and ultimate absorption amount of fresh concrete still have room for development. Increasing the $CO_2$ absorption efficiency of fresh concrete more effectively is thus the key focus of this study.

In this study, an ultrasonic vibration device is introduced into the original $CO_2$ absorption device [13,14]. The $CO_2$ AR and absorption amount of fresh cement paste under ultrasonic vibrating agitation as well as the influences of ultrasonic vibration on the fluidity and porosity of $CO_2$ absorption are discussed through the unique ultrasonic performances. The collaborative action mechanism of fresh cement paste under ultrasonic vibrating agitation and $CO_2$ absorption was analysed through SEM observation. The variation laws of the internal microstructure after $CO_2$ absorption of fresh cement paste for different amounts are observed [15–17]. On this basis, the variation mechanism of basic performances of cement-based materials as a response to ultrasonic vibration is disclosed and the influences of cement flocculates on the crystal size of solution under ultrasonic waves were are analysed thoroughly [18–20].

## 2. Materials and Methods

### 2.1. Raw Materials

The P·O 42.5 cement, which is produced by the Xuzhou Zhonglian Cement Group, was chosen for this study. The mean grain size, density, standard-consistency water need, fineness (0.08 mm square hole sieve), and specific surface area were 14.813% Xav (μm), 3.14 g/cm$^3$, 28.1%, 1.02%, and 3300 cm$^2$/g, respectively. The specific chemical composition and mineral composition are listed in Tables 1 and 2. In this experiment, $CO_2$ is the high-purity $CO_2$ which is produced by a special gas plant in Xuzhou, the purity of which is $\geq$99.5%. The sand used in this experiment was the ISO cement test standard sand and the water used was tap water.

**Table 1.** Chemical composition of cement.

| Chemical Composition | $SiO_2$ | $Al_2O_3$ | $Fe_2O_3$ | CaO | MgO | f-CaO | Loss |
|---|---|---|---|---|---|---|---|
| Content (%) | 22.1 | 5.34 | 3.44 | 65.33 | 2.11 | 0.39 | 0.13 |

**Table 2.** Mineral composition of clinkers.

| Components | $C_3S$ | $C_2S$ | $C_3A$ | $C_4AF$ |
|---|---|---|---|---|
| Proportions (%) | 54.04 | 22.84 | 8.39 | 10.42 |

### 2.2. Reconstruction of the $CO_2$ Absorption Device

Two $CO_2$ absorption devices were manufactured to discuss the influences of ultrasonic agitation on the $CO_2$ AR and the ultimate absorption amount of fresh cement paste. One is a mechanical agitation device (Figure 1a) and the other is an ultrasonic vibrating agitation tank. The manufacturing process is introduced as follows. Firstly, the ultrasonic vibrating agitation tester with different ultrasonic vibrational frequencies is manufactured by connecting the ultrasonic vibrator into the original $CO_2$ absorption device. Next, a transducer, ultrasonic power supply, and qualified amplitude transformer are designed and selected according to the requirements of the vibration system on amplitude and vibration frequency. Finally, the ultrasonic power supply, transducer, amplitude transformer, and agitation tank are connected (Figure 1b).

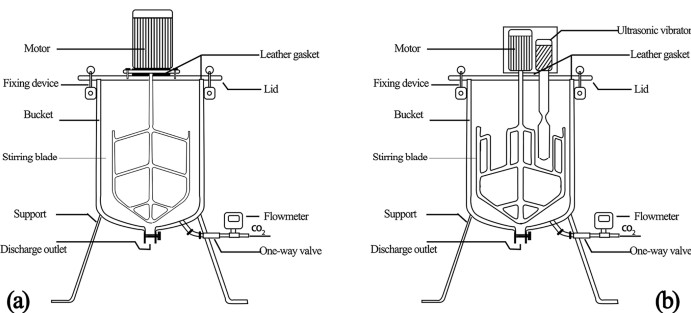

**Figure 1.** Stirring device: (**a**) Mechanical agitation device, (**b**) ultrasonic vibrating agitation device. Reprinted with permission from ref. [21]. Copyright 2021 Elsevier.

### 2.3. Fresh Cement Paste Preparation for $CO_2$ Absorption

Cement paste with a water-cement ratio of 0.5 was prepared. The stirring rate was set $210 \pm 5$ r/min. After the paste was stirred evenly, the ultrasonic vibrator and $CO_2$ flow valve were opened and $CO_2$ was supplied while stirring. The $CO_2$ AR and absorption amount of cement paste were measured by a $CO_2$ flowmeter. In this way, cement paste that meets the required $CO_2$ AA was prepared.

### 2.4. Determination of the Optimum Ultrasonic Frequency

Ultrasonic waves are acoustic waves with frequencies higher than 20 kHz, usually ranging between 20–100 kHz. The cavitation effect of an ultrasonic wave is related to its frequency. The ultrasonic frequency is negatively related to the cavitation effect of an ultrasonic wave. To reach a better ultrasonic cavitation effect, this research group developed three ultrasonic agitation devices with low ultrasonic frequencies of 20, 28, and 40 kHz [22–24].

The specimens were divided into four groups: A1, A2, A3, and A4. Groups A1, A2, and A3 were moulded using ultrasonic vibrating agitation. The ultrasonic frequencies of these three groups were 20, 28, and 40 kHz, respectively. Group A4 was used as a control group and moulded by mechanical agitation. The $CO_2$ AR and $CO_2$ AA were measured every 5 s to study the influences of ultrasonic frequency on real-time changes of the $CO_2$ AR and $CO_2$ AA. On this basis, the optimal ultrasonic frequency could be determined.

### 2.5. Experimental Procedure

#### 2.5.1. Specimen Grouping

The test was divided into Groups B and C. Group B used ultrasonic vibrating agitation and Group C was moulded by mechanical agitation as a control group. The corresponding $CO_2$ AA were 0%, 0.44%, 0.88%, 1.32%, 1.76%, and 2.20%, respectively. Cement pastes in Group B were numbered as B1, B2, B3, B4, B5, and B6. Cement pastes in Group C were C1, C2, C3, C4, C5, and C6. The fluidity, mechanical properties, pore structures, and microstructures of cement pastes in Groups B and C were tested to study the influences of ultrasonic vibrating agitation on the $CO_2$ absorption of cement pastes.

#### 2.5.2. Fluidity of Cement Paste after $CO_2$ Absorption

The fluidity of Groups B and C was tested according to the Cement and Water Reducing Agent Compatibility Test Method (JC/T1083-2008) [25]. First, we poured the cement paste after $CO_2$ absorption into the truncated cone and scraped flat. Then, we lifted the truncated cone in the vertical direction and let the paste flow. Thirty seconds later, the maximum diameters of the paste were measured in two directions perpendicular to each other. Finally, the fluidity of the cement paste is the average of the two maximum diameters.

#### 2.5.3. Pore structure of Hardened Cement Paste after $CO_2$ Absorption

The $40 \times 40 \times 160$ mm standard cement mortar specimens were prepared using fresh cement pastes of Groups B and C according to Method of Testing Cements-Determination

of Strength (GB/T17671-1999) [26–28]. All specimen moulds were removed after curing for 24 h, followed by standard curing to the regulated age under $20 \pm 2$ °C and humidity > 95%. At this moment, the compressive strength at 3, 7, and 29 d were determined by a LS80-65–160 hydraulic compression tester to discuss the influences of ultrasonic vibration on the mechanical properties of fresh cement paste after $CO_2$ absorption [29–31].

### 2.5.4. Mechanical Properties of Hardened Cement Paste after $CO_2$ Absorption

Cement pastes of Groups B and C were filled with cubic test moulds in a size of $40 \times 40 \times 40$ mm. The cement paste was made compact by vibration. Then, 24 h later, the mould was removed and specimens were cured under standard conditions of $20 \pm 2$ °C and humidity > 95% to the regulated age, followed by 24 h of drying in an oven at a temperature of 120 °C. Specimens were crushed into blocks or cylinders. Pore structural distribution and porosity after 28 d were determined by mercury intrusion porosimeter, a PoreMaster33 with a testing range of 3.5 nm to 400 μm. The effects of ultrasonic vibration on the porosity of cement paste after $CO_2$ absorption are discussed [32–34].

### 2.5.5. Characterization of the Hydration Products in Cement Paste after $CO_2$ Absorption

The fresh cement paste of Groups B and C was put into cubic test moulds of $40 \times 40 \times 40$ mm dimension. The cement paste was then compacted by vibration and cured under standard conditions for 12 h to prepare the cement paste samples. At this time, the strength of the cement paste sample is first established, which is conducive to sample preparation. Furthermore, the hydration reaction in the cement is in its initial stages and the internal structure is relatively loose, which makes it an ideal time to observe the products produced by the reaction of cement hydration products with $CO_2$.

The cement paste samples were made into 1-mm-thick pieces, which were then immersed in absolute ethanol for 48 h to prevent the hydration of the cement. The pieces were dried in a constant temperature blast oven at 65 °C for 24 h and then placed in an ion sputtering apparatus for surface gold spraying. Then, scanning electron microscopy (SEM) and energy spectrum analysis were performed with a scanning electron microscope Quanta 250.

## 3. Results

### *3.1. Effects of Ultrasonic Frequency on the $CO_2$ Ultimate Absorption Amount and AR*

3.1.1. Effects of Ultrasonic Frequency on the $CO_2$ Ultimate Absorption Amount

The effects of ultrasonic vibration on the $CO_2$ ultimate absorption amount are shown in Figure 2. It can be seen that under mechanical agitation, the $CO_2$ AA in the cement paste increased gradually. When the $CO_2$ absorption reached 5, 10, 15, and 20 s, the $CO_2$ AA were 0.77%, 1.35%, 1.87%, 2.15%, and 2.47% of the cement mass, respectively. After an absorption time of 35 s, the cement paste could not further absorb $CO_2$ due to thickening and lost fluidity. At this point, the $CO_2$ AA of the cement paste reached 2.64% of the ultimate absorption amount.

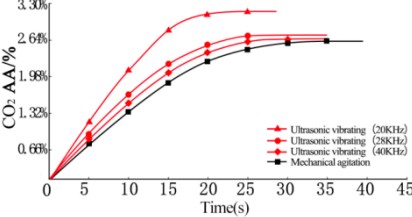

**Figure 2.** Effects of ultrasonic vibration on the $CO_2$ ultimate absorption amount.

Compared with mechanical agitation, the $CO_2$ AR of fresh cement paste is significantly increased under ultrasonic vibrating agitation. At 5, 10, 15, 20, and 25 s, the $CO_2$ AA of fresh cement pastes under the fixed ultrasonic frequency of 40 kHz were 0.85%, 1.41%, 2.07%, 2.42%, and 2.61% of cement mass, respectively. These are 10.4%, 4.4%, 10.7%, 12.6%,

and 5.7% higher than those under mechanical agitation at the same absorption times. The $CO_2$ AA reached a maximum at 30 s, approximately 2.66% of the ultimate absorption amount. Compared with mechanical agitation, the $CO_2$ ultimate absorption amount of fresh cement pastes under ultrasonic vibrating agitation is significantly increased with the shortening of absorption time.

With the reduction of ultrasonic frequency, the $CO_2$ ultimate absorption amount of cement paste is increased to some extent. At 5, 10, 15, and 20 s, the $CO_2$ AA of fresh cement pastes at the fixed ultrasonic frequency of 28 kHz were 0.94%, 1.65%, 2.2%, and 2.53% of the cement mass. These increased to 1.19%, 2.09%, 2.84%, and 3.12% by decreasing the ultrasonic frequency to 20 kHz. At an ultrasonic frequency of 28 kHz, the $CO_2$ AA of fresh cement pastes reached 2.73% of the ultimate absorption amount at 25 s. At an ultrasonic frequency of 20 kHz, the $CO_2$ AA of fresh cement pastes reached 3.17% of the ultimate absorption amount at 25 s.

### 3.1.2. Effects of Ultrasonic Vibration on $CO_2$ AR

The effects of ultrasonic vibration on $CO_2$ AR are shown in Figure 3. It can be seen that under mechanical agitation, the $CO_2$ AR was 0.063, 0.061, 0.058, 0.053, 0.046, and 0.035 $\nu(\%/s)$ at the absorption times of 5, 10, 15, 20, 25, and 30 s, respectively. By increasing the absorption time, the cement paste gradually thickens causing the $CO_2$ AR to decrease accordingly. The cement paste changed from a fluid to pasty fluid gradually after 30 s. At 35 s, the $CO_2$ absorption of the paste reached saturation and the pastes could not absorb $CO_2$ any further.

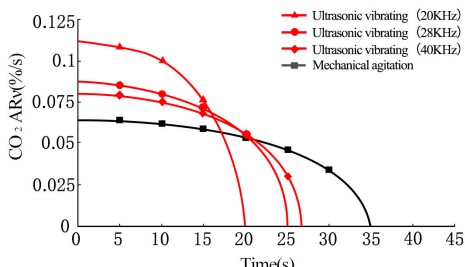

**Figure 3.** Effects of ultrasonic vibration on $CO_2$ AR.

The $CO_2$ AR of fresh cement paste increased significantly under ultrasonic vibrating agitation compared to that under mechanical agitation. At an ultrasonic frequency of 40 kHz, the $CO_2$ AR of the cement paste was 0.08, 0.075, 0.067, and 0.055 $\nu(\%/s)$ at the absorption times of 5, 10, 15, and 20 s, respectively, which are 27%, 23%, 15.5%, and 3.8% higher compared to those under mechanical agitation. After 20 s, the cement paste changes from fluid to pasty fluid and the $CO_2$ absorption of paste reaches saturation at 27 s.

At an ultrasonic frequency of 28 KHz, the $CO_2$ AR of cement paste is 0.085, 0.08, 0.072, and 0.054 $\nu(\%/s)$ at the absorption times of 5, 10, 15, and 20 s, respectively. The $CO_2$ AR of cement paste increases to some extent with the decrease of ultrasonic frequency. At an ultrasonic frequency of 20 kHz, the $CO_2$ AR of cement paste is 0.108, 0.1, and 0.077 $\nu(\%/s)$ at 5, 10, and 15 s respectively. The $CO_2$ AR of cement paste increases significantly.

According to the results, it can be concluded that the $CO_2$ absorption time of cement paste gradually shortens, while the $CO_2$ AR and $CO_2$ AA further increase when the ultrasonic frequency gradually decreases from 40 kHz to 28 and 20 kHz. This is due to the fact that when the ultrasonic frequency increases, the ultrasonic intensity will increase accordingly. When the increased ultrasonic intensity is excessive, there are excessive bubbles produced, which conversely increase attenuation of scattering, forming barriers of the ultrasonic wave. On the other hand, increasing of ultrasonic intensity will also lead to increasing of nonlinear attenuation, which is disadvantageous for uniform agitation. Consequently, particles of the cement paste are not well distributed. Therefore, when the

ultrasonic frequency is 20 kHz, the cavitation effect of ultrasonic agitation is best, and the corresponding $CO_2$ AR and $CO_2$ AA is highest.

### 3.2. Effects of Ultrasonic Vibration on the Fluidity of Cement Paste after $CO_2$ Absorption

3.2.1. Divergence of Cement Paste after $CO_2$ Absorption

The changes of divergence of cement paste with $CO_2$ AA are shown in Figure 4. From Figure 4a, it can be seen that under mechanical agitation, the divergence values of cement paste are 159, 140, 131, 122, 106, and 98 mm when the $CO_2$ AA are 0%, 0.44%, 0.88%, 1.32%, 1.76%, and 2.20%, respectively. Figure 4b shows that the cement paste surface looks finer and more watery after ultrasonic vibrating agitation. Under ultrasonic vibrating agitation, divergence values of the cement paste are 174, 154, 143, 133, 115, and 105 mm, respectively. These are 9.4%, 10% 9.2%, 8.5%, and 7.1% higher than those under mechanical agitation. Ultrasonic vibration can increase the divergence of cement paste effectively. The divergence of cement paste decreases gradually with the increase of $CO_2$ AA and the paste loses fluidity slowly to gradually become a pasty fluid.

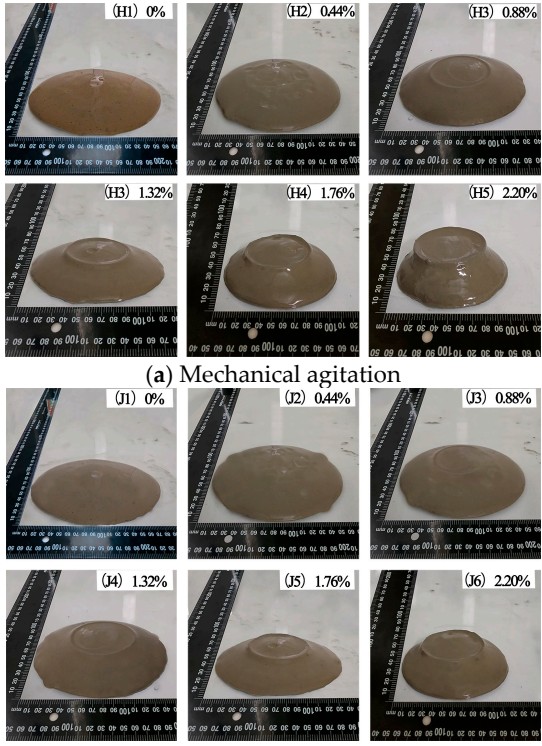

(**a**) Mechanical agitation

(**b**) Ultrasonic vibrating agitation

**Figure 4.** Changes of divergence of cement paste with $CO_2$ AA.

3.2.2. Effects of Ultrasonic Vibration on the Fluidity of Cement Paste after $CO_2$ Absorption

The effects of ultrasonic vibration on the divergence of cement paste after $CO_2$ absorption were drawn according to test results of divergence (Figure 5). Clearly, under the same $CO_2$ AA, the divergence of fresh cement paste under the ultrasonic vibrating agitation is higher than that under mechanical agitation. Moreover, the divergence of cement paste values were 174, 154, 143, 133, 115, and 105 mm when the $CO_2$ AA were 0%, 0.44%, 0.88%, 1.32%, 1.76%, and 2.20%, respectively, which decreased by 13.0%, 7.7%, 7.5%, 15.7%, and 9.5%, respectively.

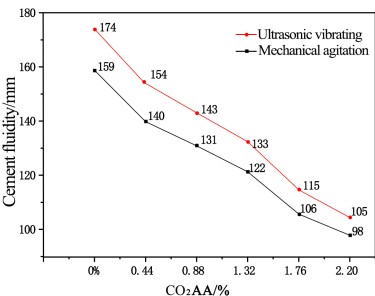

**Figure 5.** Effects of $CO_2$ AA on the fluidity of cement paste.

The above results indicate that the ultrasonically agitated fluidity of cement paste is increased compared with that mechanically agitated. Cement particles in unit volume of cement paste is significantly increased when ultrasonic agitation is adopted, which means that ultrasonic agitation is better to break the flocculation structures formed by cement particles compared with mechanical agitation. The break of flocculation structures means that there is more free water in the paste to ensure fluidity of cement paste. However, both fluidities of cement paste agitated ultrasonically and mechanically decreases with an increase in $CO_2$ AA and the decrement is significant. This is due to the fact that the structure of the cement paste gradually changes when $CO_2$ AA gradually increases, which is disadvantageous for fluidity of cement paste.

### 3.3. Effects of Ultrasonic Vibration on Pore Distribution and Porosity of Hardened Cement Paste after $CO_2$ Absorption

3.3.1. The Most Available Geometric Diameter of Pore Distribution

The differential curve of the pore diameter of hardened cement paste under $CO_2$ AA was tested by the mercury intrusion method (Figure 6). For $CO_2$ AA of 0%, 0.44%, 0.88%, 1.32%, 1.76%, and 2.20%, the most available geometric diameters in the pore distribution differential curve of hardened cement paste are 112.5, 118.5, 92.5, 83.1, 80.7, and 72.7 nm under mechanical agitation. Under ultrasonic vibrating agitation, the most available geometric diameters in the pore distribution differential curve of hardened cement paste are 78.7, 83.3, 70.5, 60.6, 49.4, and 48.2 nm, respectively, which are decreased by 30%, 29.7%, 23.8%, 27.1%, 38.8%, and 33.7% compared to those under mechanical agitation.

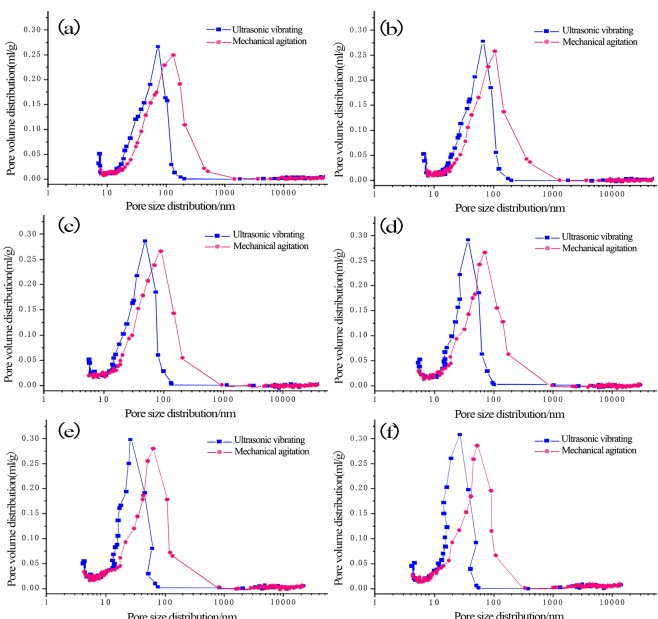

**Figure 6.** Differential curve of pore distributions of hardened cement paste under different $CO_2$ AA: (**a**) 0%; (**b**) 0.44%; (**c**) 0.88%; (**d**) 1.32%; (**e**) 1.76%; (**f**) 2.20%.

It can be obtained according to Figure 6 that the most available geometric diameter of pore distribution of the hardened carbonized cement paste with ultrasonic agitation is lower than that with mechanical agitation. When the $CO_2$ AA is respectively 0.44%, 0.88%, 1.32%, 1.76%, and 2.20%, the most available geometric diameter of pore distribution of the hardened carbonized cement paste with ultrasonic agitation gradually decreases by 15.4%, 14%, 18.5%, and 2.4% compared to that with mechanical agitation, which indicates that the most available geometric diameter of pore distribution of the hardened carbonized cement paste with ultrasonic agitation gradually decreases with an increase in $CO_2$ AA. Therefore, the most available geometric diameter of pore distribution of the hardened carbonized cement paste can effectively be reduced with ultrasonic agitation.

### 3.3.2. The Most Available Geometric Diameter of Pore Distribution

Wu Zhongwei divided the concrete pore into four types according to influences of pore size on the durability of concretes: Harmless pores (<20 nm), slightly harmful pores (20~100 nm), harmful pores (100~200 nm), and multi-harmful pores (>200 nm). Influences of ultrasonic vibration on pore distribution in the hardened cement paste after $CO_2$ absorption are shown in Figure 7. Clearly, the percentages of harmless pores are 4%, 5%, 6%, 8%, 9%, and 9% under mechanical agitation when the $CO_2$ AA increases from 0% to 2.20% at a rate of 0.44%. Meanwhile, the percentages of slightly harmful pores are 58%, 60%, 61%, 63%, 65%, and 67%, respectively. The percentages of harmful pores are 34%, 31%, 29%, 26%, 23%, and 21%, respectively. The percentages of multi-harmful pores are 4%, 4%, 4%, 3%, 3%, and 3%. Under ultrasonic vibrating agitation, the percentages of harmless pores are 6%, 6%, 8%, 9%, 10%, and 10% when the $CO_2$ AA increases from 0% to 2.20% at a rate of 0.44%. Meanwhile, the percentages of slightly harmful pores are 70%, 72%, 74%, 77%, 77%, and 78%, respectively. The percentages of harmful pores are 21%, 19%, 15%, 12%, 11%, and 10%, respectively. The percentages of multi-harmful pores are 3%, 3%, 3%, 2%, 2%, and 2%, respectively.

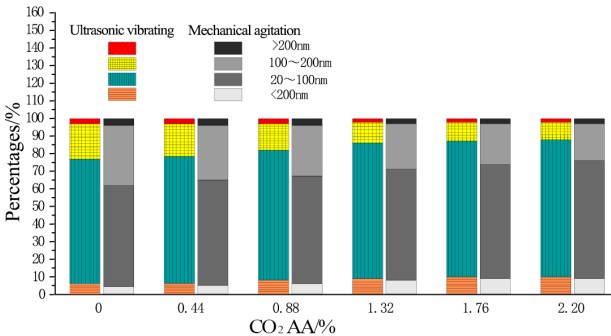

**Figure 7.** Pore distributions in hardened cement paste under different $CO_2$ AA.

According to the test results, slightly harmful pores and harmful pores change mostly in the pore distribution in hardened cement paste after $CO_2$ absorption under ultrasonic vibrating agitation. Specifically, the proportion of slightly harmful pores increases significantly, while the proportion of harmful pores decreases dramatically. When the $CO_2$ AA are 0%, 0.44%, 0.88%, 1.32%, 1.76%, and 2.20%, the percentages of slightly harmful pores are increased by 20.7%, 20%, 21.3%, 22.2%, 18.8%, and 16.4%, respectively, and the percentages of harmful pores are decreased by 38.2%, 38.7%, 48.3%, 53.8%, 52.2%, and 52.4%, respectively.

It can be obtained from Figure 7, with an increase in $CO_2$ AA, that harmless pores and slightly harmful pores of the hardened carbonized cement paste will increase, while harmful pores and multi-harmful pores will decrease. However, the increment of harmless pores and slightly harmful pores and the decrement of harmful pores and multi-harmful pores are more significant when ultrasonic agitation is adopted.

### 3.3.3. Porosity

The total porosity of hardened cement pastes after different $CO_2$ AA under ultrasonic vibrating agitation is shown in Figure 8. Clearly, the porosities of the hardened cement paste under mechanical agitation are 17.5%, 17.1%, 16%, 15.1%, 14.3%, and 13.8% when the $CO_2$ AA are 0%, 0.44%, 0.88%, 1.32%, 1.76%, and 2.20%, respectively. Under ultrasonic vibrating agitation, the porosities of the hardened cement paste are 15.6%, 15.1%, 14.6%, 13.4%, 12.1%, and 11.5%, which are 10.9%, 11.7%, 8.8%, 1.3%, 15.4%, and 16.7% lower compared to those under mechanical agitation, respectively. Moreover, the porosities of hardened cement paste decrease by 3.2%, 3.3%, 8.2%, 9.7%, and 5% when the $CO_2$ AA increases at the rate of 0%, 0.44%, 0.88%, 1.32%, 1.76%, and 2.20% continuously under ultrasonic vibrating agitation.

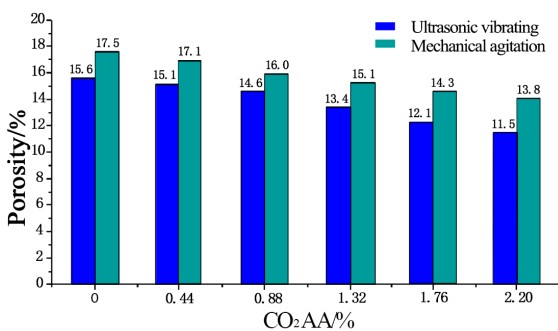

**Figure 8.** Porosity of hardened cement paste after $CO_2$ absorption.

According to the above analyses, conclusions can be drawn that the porosity of the hardened carbonized cement paste whether mechanically agitated or ultrasonically agitated decreases when $CO_2$ AA increases. However, the porosity decrement is more obvious when ultrasonic agitation is applied. Reasons for the significant porosity decrement when ultrasonic agitation is applied lies in two aspects. On the one hand, cement particles can be minimized by ultrasonic waves, which is advantageous for the hydration degree of cement particles. On the other hand, the flocculation structures formed by cement particles can be broken by ultrasonic agitation, which is advantageous for cement particles to get accesses to water to improve the hydration degree. The improved hydration degree of cement particles indicates that more hydration products of C-S-H and CH are formed, which improves the density of the hardened cement paste, thus reducing porosity of the hardened cement paste.

### 3.3.4. Mean Pore Size

The mean pore sizes of hardened cement paste after different $CO_2$ AA under ultrasonic vibrating agitation are shown in Figure 9. It can be seen that under mechanical agitation, the mean pore sizes of hardened cement paste are 88, 79, 75, 68, 65, and 60 nm when $CO_2$ AA are 0%, 0.44%, 0.88%, 1.32%, 1.76%, and 2.20%, respectively. Meanwhile, the mean pore sizes of hardened cement paste are 80, 73, 65, 56, 50, and 45 nm under ultrasonic vibrating agitation, which are decreased by 9.1%, 7.6%, 13.3%, 17.6%, 23.1%, and 25% compared to those under mechanical agitation. Furthermore, the mean pore sizes of hardened cement paste under ultrasonic vibrating agitation decrease by 8.8%, 23.3%, 13.8%, 10.7%, and 10% when the $CO_2$ AA increases at the rate of 0%, 0.44%, 0.88%, 1.32%, 1.76%, and 2.20%, continuously.

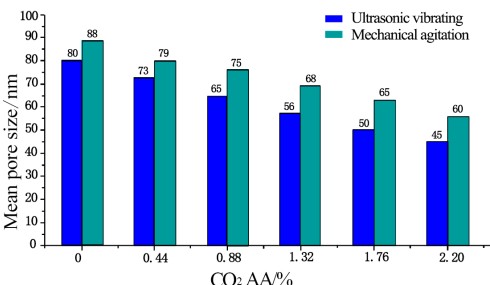

**Figure 9.** Mean pore size.

According to the above analyses, it can be obtained that both mechanical and ultrasonic agitation is helpful to minimize the mean pore size and that the mean pore size gradually decreases with an increase in $CO_2$ AA. However, the decrement in mean pore size when ultrasonic agitation is applied is more obvious. This is due to the fact that the products of $CaCO_3$ crystals are helpful to minimize pore sizes and optimize pore size distribution of the hardened carbonized cement paste.

### 3.4. Effects of Ultrasonic Vibration on the Mechanical Properties of Cement-Based Materials after $CO_2$ Absorption

#### 3.4.1. Compressive Strength

The effects of ultrasonic vibration on the compressive strength of cement-based materials after $CO_2$ absorption are shown in Figure 10. At the age of 3 d, the $CO_2$ AA are 0%, 0.44%, 0.88%, 1.32%, 1.76%, and 2.20% with a corresponding compressive strength of the hardened cement paste of 5.9, 5.4, 5.7, 5.2, 5.6, and 5.3 MPa under mechanical agitation. According to the data, the compressive strength may increase and decrease with the increase of $CO_2$ AA, but the fluctuation amplitude is not very large. At the age of 7 and 28 d, the compressive strength of hardened cement paste is increased to some extent. At 7 d, the compressive strengths are 6.9, 6.7, 6, 6.1, 6.5, and 6.5 MPa, respectively. The variation law of the compressive strength firstly decreases slowly but decreases significantly when the $CO_2$ AA is 0.88%. Subsequently, it increases with the increase of $CO_2$ AA.

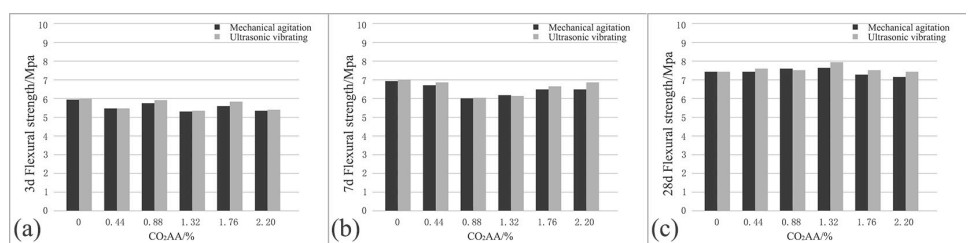

**Figure 10.** Effects of ultrasonic vibration on the compressive strength of cement-based materials after $CO_2$ absorption: (**a**) 3 d; (**b**) 7 d; (**c**) 28 d.

At the age of 28 d, the compressive strengths of the hardened cement paste are 7.4, 7.4, 7.6, 7.6, 7.2, and 7.1 MPa, respectively. In view of the data, the compressive strength changes stably and even presents a slowly increasing trend in the early stages of $CO_2$ absorption as the absorption amount increases. However, the compressive strength decreases to a small extent with the increase of absorption amount, though the variation amplitude is not evident. At different ages, the compressive strength of fresh cement paste does not increase significantly with an increase of $CO_2$ AA. It does change to some extent in the middle, however, it does not change greatly with $CO_2$ AA.

When the $CO_2$ AA are 0%, 0.44%, 0.88%, 1.32%, 1.76%, and 2.20%, under ultrasonic vibrating agitation, the compressive strengths of the hardened cement paste at 3 d are 6, 5.5, 5.9, 5.3, 5.8, and 5.4 MPa, respectively. In view of the data, the variation law of compressive strength is very close to that under mechanical agitation. At 7 d, the compressive strengths of hardened cement paste are 6.9, 6.7, 6, 6.1, 6.5, and 6.5 MPa, respectively, which then

increases to 7.4, 7.6, 7.5, 7.9, 7.5, and 7.4 MPa at 28 d, respectively. Meanwhile, the compressive strength at 7 and 28 d under ultrasonic vibrating agitation approaches the results under mechanical agitation as the $CO_2$ AA increases.

The variation laws of compressive strength under mechanical agitation and ultrasonic vibrating agitation are essentially identical. The compressive strength of hardened cement paste under ultrasonic vibrating agitation is increased to some extent compared with those after equal $CO_2$ AA at the same age under mechanical agitation. With the increase of $CO_2$ AA, the compressive strength changes slightly, indicating that ultrasonic vibration influences the compressive strength of cement paste after $CO_2$ absorption slightly.

### 3.4.2. Compressive Strength

The effects of ultrasonic vibration on the comprehensive strength of cement-based materials after $CO_2$ absorption are shown in Figure 11. It can be seen that when the $CO_2$ AA are 0%, 0.44%, 0.88%, 1.32%, 1.76%, and 2.20%, under mechanical agitation, the compressive strength of hardened cement paste at 3 d are 24.9, 24.4, 24.9, 24, 25, and 24.9 MPa, respectively. In view of the data, the compressive strength of cement paste after $CO_2$ absorption within the curing stage changes slightly. Even though it fluctuates to some extent, the fluctuation range is very small. At 7 d, the compressive strengths of hardened cement paste are 32.6, 33.4, 32.1, 32.8, 33.8, and 33.8 MPa, respectively. At 28 d, the compressive strengths of hardened cement paste are 45, 44.1, 45.8, 47, 46.5, and 46.7 MPa, respectively. In a word, the compressive strength of hardened cement paste after $CO_2$ absorption does not change with the increase of $CO_2$ AA as the curing age increases.

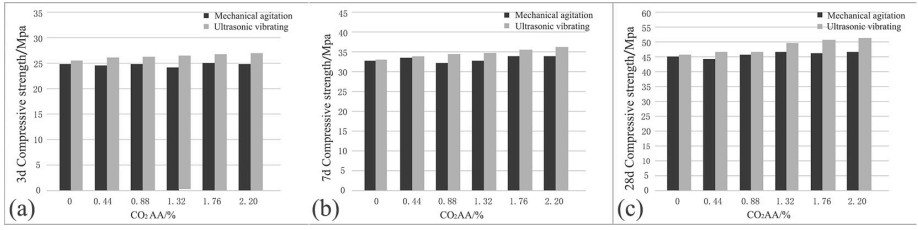

**Figure 11.** Effects of ultrasonic vibration on comprehensive strength of cement-based materials after $CO_2$ absorption: (**a**) 3 d; (**b**) 7 d; (**c**) 28 d.

When the $CO_2$ AA are 0%, 0.44%, 0.88%, 1.32%, 1.76%, and 2.20%, under ultrasonic vibrating agitation, the compressive strengths of hardened cement paste at 3 d are 25.5, 26.4, 26.6, 26.8, 27, and 27.1 MPa, respectively. The compressive strength under ultrasonic vibrating agitation is increased to some extent compared to that of mechanical agitation. Moreover, the compressive strength is positively related to the $CO_2$ AA. At 7 d, the compressive strengths of hardened cement paste are 33, 34, 34.5, 34.8, 35.5, and 36.5 MPa. At 28 d, the compressive strengths of hardened cement paste are 45.4, 47, 47, 49.5, 51, and 51.7 MPa.

According to the above experimental results, it can be obtained that the increment in compressive strength of the hardened carbonized cement paste is more significant when ultrasonic agitation is applied compared with that when mechanical agitation is applied. Under ultrasonic agitation, cement particles are broken to smaller ones and flocculation structures formed by cement particles are broken by ultrasonic waves, which can promote cement hydration to produce more C-S-H. Therefore, the compressive strength is significantly increased by applying ultrasonic.

## 4. Microstructural Analysis of Cement Paste after $CO_2$ Absorption under Ultrasonic Vibration

### 4.1. Mechanical Agitation Molding

SEM images of the mechanical agitation moulded specimens of cement paste after 10,000 amplifications are shown in Figure 12. Obviously, the microstructure of cement paste without $CO_2$ absorption in the early hardening stage shows relatively sparse cement particle

distribution and gelatinization structures in the paste compared with the microstructure when $CO_2$ AA is 0.44%. Moreover, a significant number of pores in the paste and hydration products are scattered around, revealing poor structural integrity (Figure 12a). In contrast, cement paste with a $CO_2$ AA of 0.44% shows few $CaCO_3$ crystals under SEM. After amplification, there are few $CaCO_3$ needle-like crystal whiskers in the microstructure which penetrate in gel substances (Figure 12b) [35]. However, no crystal whiskers are found in cement paste that has not absorbed $CO_2$.

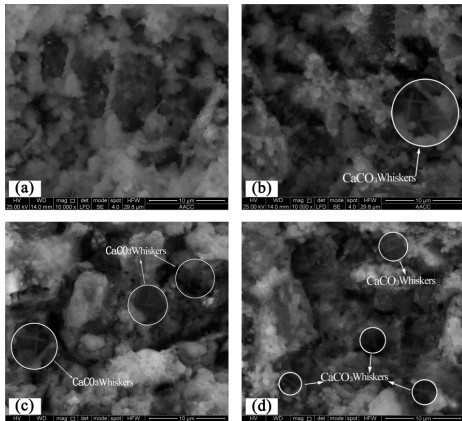

**Figure 12.** Microstructures of mechanical agitation moulded cement paste in the early hardening stage: (**a**) 0%; (**b**) 0.44%; (**c**) 0.88%; (**d**) 1.32%.

When the $CO_2$ AA is increased to 0.88%, hydration products are distributed more extensively in the paste. The $CaCO_3$ crystals of hydration products and $CaCO_3$ needle-like crystal whiskers are increased, while the number of pores is decreased (Figure 13c). When the $CO_2$ AA increases to 1.32%, the internal structure becomes more compact in the early stage of $CO_2$ absorption of the paste. Additionally, $CaCO_3$ needle-like crystal whiskers of hydration products are increased, which interweave both vertically and horizontally (Figure 12d).

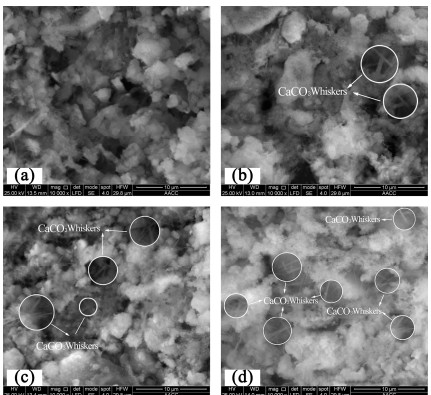

**Figure 13.** Microstructures of ultrasonic agitation moulded specimens of cement paste in the early hardening stage: (**a**) 0%; (**b**) 0.44%; (**c**) 0.88%; (**d**) 1.32%.

### 4.2. Ultrasonic Vibrating Agitation Moulding

SEM images of ultrasonic vibrating agitation moulded specimens of cement paste after 10,000 amplifications are shown in Figure 13. It can be seen that under ultrasonic vibrating agitation, the microstructure of cement pastes without $CO_2$ absorption in the early hardening stage shows relatively uniform distributions of cement particles and gel structures compared with the microstructure when the $CO_2$ AA is 0.44%. Moreover, there are no big pores, and the pores are in a relatively uniform distribution accompanied by a

reduction of distances among flocculating constituents (Figure 13a). In contrast, the cement paste with a $CO_2$ AA of 0.44% shows some $CaCO_3$ crystals and the existence of some $CaCO_3$ needle-like crystal whiskers penetrating in the gel substances, which are similar to those of mechanical agitation moulded specimens (Figure 13b).

When the $CO_2$ AA is 0.88%, $CO_2$ and $Ca^{2+}$ react continuously as $CO_2$ is supplied continuously, accelerating the hydration process of cement and producing hydration products continuously. As a result, pores in the paste are filled continuously, effectively decreasing the quantity and diameter of pores at the same time as increasing the volume of flocculating constituents and quantity of $CaCO_3$ needle-like crystal whiskers (Figure 13c). When the $CO_2$ AA is 1.32%, the flocculating constituents increase in uniform distribution and the number of pores is decreased significantly in view of the microstructure. Furthermore, the $CaCO_3$ whiskers continue to increase and interact to form cubic network structures (Figure 13d). Compared with mechanical agitation, the distributions of cement particles and hydration products under ultrasonic vibration are more uniform, and the number of pores decreases, whilst the production of $CaCO_3$ needle-like crystal whiskers increases. Hence, cement becomes more compact in the early hardening stage [36].

### 4.3. Energy Dispersive Spectrum (EDS) Analysis

In order to better understand the content changes of the elements of the paste after $CO_2$ absorption by adopting ultrasonic vibration technology, EDS analysis was conducted with the needle-like production in cement paste without $CO_2$ absorption, respectively and the cement paste with 1.32% $CO_2$ absorption by mechanical agitation, as well as that with 1.32% $CO_2$ absorption by ultrasonic agitation, as illustrated in Figure 14a–c. The contents of each element are shown in Table 3.

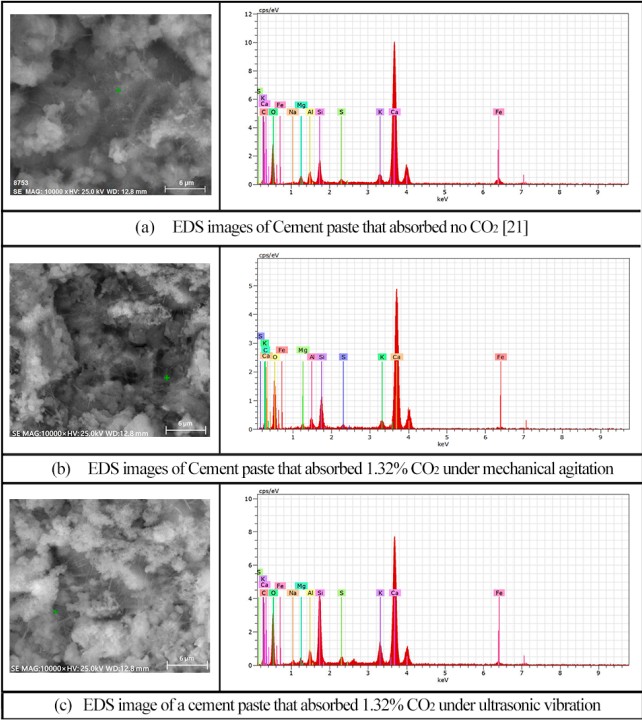

(a) EDS images of Cement paste that absorbed no $CO_2$ [21]

(b) EDS images of Cement paste that absorbed 1.32% $CO_2$ under mechanical agitation

(c) EDS image of a cement paste that absorbed 1.32% $CO_2$ under ultrasonic vibration

**Figure 14.** EDS analysis.

**Table 3.** EDS analysis results.

| Element | Molar Percentage | | |
| --- | --- | --- | --- |
| | Figure 14a | Figure 14b | Figure 14c |
| C·K | 11.38 | 14.23 | 16.09 |
| O·K | 68.92 | 63.22 | 71.35 |
| Si·K | 2.47 | 2.81 | 2.62 |
| Ca·K | 11.66 | 12.01 | 13.11 |
| Al·K | 2.18 | 1.56 | 1.24 |
| Mg·K | 1.33 | 0.72 | 1.52 |
| K·K | 1.24 | 0.83 | 0.98 |
| Fe·K | 0.33 | 0.44 | 0.27 |
| S·K | 0.50 | 0.38 | 0.33 |

According to Table 3, the main elements of the measure points in Figure 14a–c are C, O, and Ca, respectively. Meanwhile, there are also some Si and Al elements. It can be obtained from Table 3 that the contents of C, Ca, and Si elements are increased, while that of Al element is obviously reduced when $CO_2$ is absorbed by the cement paste under mechanical agitation. However, under ultrasonic agitation, the contents of C and Ca elements are increased more obviously and that of Al element continues to decrease.

According to the above results, it can be concluded that with the absorption of $CO_2$, the content of C element of the cement paste is increased and there are $CaCO_3$ crystals in the cement paste. Moreover, due to the "cavitation effect" of ultrasonic agitation, $CO_2$ is effectively dispersed in cement paste, leading to more $CaCO_3$ crystals being produced by adopting ultrasonic agitation. Therefore, cement paste is able to absorb more $CO_2$, thus forming more $CaCO_3$, when ultrasonic agitation is applied compared with that when mechanical agitation is applied.

## 5. Mechanism Analysis

### 5.1. Hydration Mechanism of Cement Paste under Mechanical Agitation

#### 5.1.1. Cement Paste without $CO_2$ Absorption

The hydration process of cement paste without $CO_2$ absorption is shown in Figure 15. Clearly, a gel film that is composed of calcium silicate hydrate (C-S-H) gel and calcium hydroxide (CH) crystals is formed on the surface of cement particles in the early hydration stage. With the continuous increase of hydration time, the gel film thickens day by day due to the increasing gels. Meanwhile, the cement begins to harden slowly until finishing the hydration of the cement.

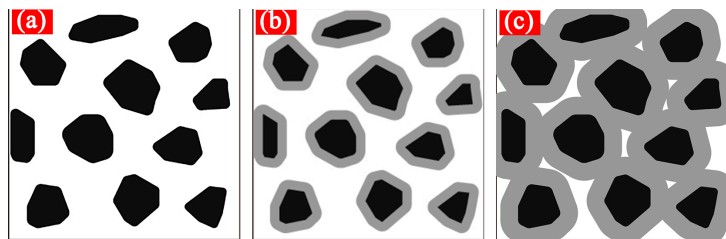

**Figure 15.** Hydration mechanism of fresh cement paste under mechanical agitation: (**a**) Initial state of fresh cement paste; (**b**) early hydration stage of cement paste; (**c**) late hydration stage of cement paste. (Note: ⬣ cement particles; ⬠ hydration products of flocculating gel).

#### 5.1.2. Cement Paste after $CO_2$ Absorption under Mechanical Agitation

The hydration process of cement paste after $CO_2$ absorption under mechanical agitation is shown in Figure 16. It can be seen that under mechanical agitation, $CO_2$ gases scatter uniformly in the cement paste and firstly dissolve into $H_2CO_3$. $H_2CO_3$ reacts with $Ca(OH)_2$ which is precipitated from the initial hydration of cement to produce flocculent

$CaCO_3$ crystals which adhere to the surface of cement particles (Figure 16a). With the increase of $CO_2$ AA, the $CaCO_3$ gel layer on the cement particle surface thickens and the cement paste viscosifies gradually, decreasing the fluidity accordingly [37].

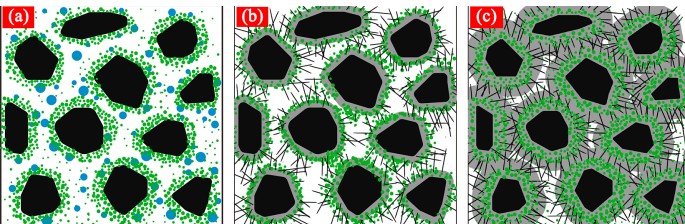

**Figure 16.** Hydration mechanism of fresh cement paste after $CO_2$ supply under mechanical agitation: (**a**) Carbonization reaction stage of cement paste after $CO_2$ supply; (**b**) early hydration stage of cement paste after $CO_2$ supply; (**c**) late hydration stage of cement paste after $CO_2$ supply. (Note: Cement particle; the supplied $CO_2$ gases; flocculent gel hydration products; $CaCO_3$ crystals; $CaCO_3$ needle-like crystal whiskers).

The $CaCO_3$ gel layer on the cement particle surface has a loosened structure and then the cement particles begin to hydrate gradually. Calcium metasilicate (C-S-H) gel and $CaCO_3$ gel which are the hydration products of cement combine on the cement particle surface into a gel layer. Some flocculent $CaCO_3$ gel crystals penetrate the C-S-H and flocculent $CaCO_3$ gels as needle-like whiskers, forming a skeletal network (Figure 16b).

In the late hydration stage of cement, hydration products on the cement particle surface increase significantly and the wrapper thickens accordingly. Moreover, $CaCO_3$ crystals are wrapped in the hydration products. Meanwhile, the $CaCO_3$ needle-like crystal whiskers are still in the paste to form an effective network structure (Figure 16c), which improves the gelling performance among cement particles.

### 5.2. Hydration Mechanism of Cement Paste under Ultrasonic Agitation

The hydration mechanism of cement paste under ultrasonic agitation is shown in Figure 17. When $CO_2$ gases are supplied under ultrasonic vibration, they firstly react with $Ca(OH)_2$ which is precipitated from the initial hydration to produce flocculent $CaCO_3$ crystals. The solid surfaces suspended in the liquid are damaged dramatically due to the "cavitation effect" of the ultrasonic wave [38]. When an ultrasonic wave radiates and spreads throughout the paste, it will produce numerous small bubbles in the paste, which will break continuously. More than 1000 instant high pressure regions can be produced at the rupture of bubbles [39–41]. The rupture explosions in the series will release a considerable amount of energy to cause great impacts on the surrounding areas. On the one hand, this causes continuous impacts on the cement particle surfaces which have not hydrated completely in the cement paste to make the gel hydration products on the surface peel off quickly, thus getting new cement particles. On the other hand, $CaCO_3$ flocculating constituents are cut effectively under the ultrasonic wave effect and the abundant "nano" crystals are produced scattered among cement particles (Figure 17a).

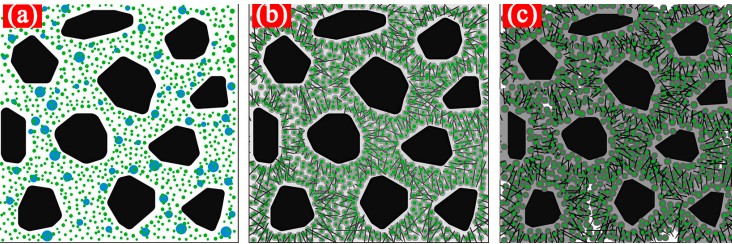

**Figure 17.** Hydration mechanism of cement paste under ultrasonic agitation: (**a**) Carbonization reaction stage of cement paste after $CO_2$ supply under ultrasonic agitation; (**b**) early hydration stage of cement paste after $CO_2$ supply under ultrasonic agitation; (**c**) late hydration stage of cement paste after $CO_2$ supply under ultrasonic agitation. (Note: ⬛ Cement particle; 🔵 the supplied $CO_2$ gases; ⬠ flocculent gel hydration products; ∴ $CaCO_3$ crystals; 𝌆 $CaCO_3$ needle-like crystal whiskers).

In the early reaction stage of $CO_2$, the $CaCO_3$ gel layer on the cement particle surface has a relatively loose structure. When $CO_2$ gases are consumed completely, the cement particles begin to hydrate gradually, producing hydration products continuously including "nano" $CaCO_3$ crystals and a certain quantity of $CaCO_3$ whiskers. Due to the "cavitation effect" of ultrasonic vibration, the "nano" $CaCO_3$ crystals cannot adhere to the cement particle surface. Since the surfaces of hydration products from the reaction of cement particles and water are positively or negatively charged and the crystal nucleus has some adsorption capacity, some hydration products can adhere onto "nano" $CaCO_3$ crystal surfaces. These "nano" $CaCO_3$ flocculating constituents are effectively filled in the spaces among the cement particles. Moreover, $CaCO_3$ whiskers depend on flocculating constituents and form a dense networked structure (Figure 17b).

In the late hydration stage of the cement, hydration products on the cement particle surfaces increase, thus thickening the wrapper on the cement particle surface. The "nano" $CaCO_3$ crystals become wrapped by hydration products of the cement which are produced continuously. Consequently, flocculating constituents in the spaces of cement particles become tighter. Moreover, the microstructure of the cement after hydration seems to have better gelling performances and a more compact structure than that under mechanical agitation since the $CaCO_3$ needle-like crystal whiskers interweave between the cement particles (Figure 17c).

## 6. Conclusions

The results of our previous investigation of $CO_2$ absorption performance of fresh cement paste indicate that the $CO_2$ absorption rate and the ultimate absorption amount of fresh cement paste can effectively be increased. However, the mixing process of concrete is relatively very short in practice. Therefore, the rapid and significant $CO_2$ absorption by concrete cannot be realized only by reasonably setting the mechanical mixing rate and water-cement ratio.

As a consequence, a new method of ultrasonic agitation is applied to facilitate the $CO_2$ absorption performance of cement paste in this investigation. Meanwhile, the $CO_2$ absorption rate and the ultimate absorption amount of cement paste under ultrasonic agitation is thoroughly studied. The results indicate that by applying ultrasonic agitation, the $CO_2$ absorption rate and the ultimate absorption amount of cement paste can be effectively increased. Once the carbonized cement paste is hardened, the hardened cement paste increases the compressive strength and decreases the porosity and pore sizes, indicating the refined pore structure of the hardened cement paste.

Due to the limited investigation on $CO_2$ absorption of cement paste, there are few literatures reporting about it. Moreover, due to the limitation of the experimental device, only

some critical problems are studied in this investigation regarding the $CO_2$ absorption of cement paste. However, there are still many problems that require thorough investigations.

**Author Contributions:** Study design, Y.J.; conduct of the study, L.L. and F.G.; conceptualization, Z.M.; data collection, Z.X.; methodology, L.L.; writing—original draft preparation, L.L. and Y.J. All authors have read and agreed to the published version of the manuscript.

**Funding:** This work was funded by the National Natural Science Foundation of China (51972337) and National Key R&D Program of China (2019YFE0118500, 2019YFC1904304).

**Institutional Review Board Statement:** Not applicable.

**Informed Consent Statement:** Not applicable.

**Data Availability Statement:** Not applicable.

**Acknowledgments:** The authors would like to thank the technical staff of the Jiangsu Key Laboratory Environmental Impact and Structural Safety in Engineering for their technical support.

**Conflicts of Interest:** The authors declare no conflict of interest.

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
