# Peer review of "Study on the Effects of Ultrasonic Agitation on CO2 Adsorption Efficiency Improvement of Cement Paste"

_applsci, doi:10.3390/app11156877_

Round 1

Reviewer 1 Report

In my opinion, this is an interesting document because the authors propose a novel topic related to the study of the effects of ultrasonic agitation for the absorption of CO2 in cement paste.

The authors adequately describe the experimental methods, but when they explain the results, they do not include correlations between variables. 

Apparently, it is possible to obtain certain interesting correlations between the results obtained.

The section 5 is very interesting, but it could provide some information regarding the chemical composition from SEM analysis, including the components in the transition zones between the paste and the aggregate.

The conclusions section should be completed with a general conclusion of the paper.

Author Response

Reviewer # 1:

In my opinion, this is an interesting document because the authors propose a novel topic related to the study of the effects of ultrasonic agitation for the absorption of CO2 in cement paste.

Point 1:The authors adequately describe the experimental methods, but when they explain the results, they do not include correlations between variables. Apparently, it is possible to obtain certain interesting correlations between the results obtained.

Response 1:Thanks for your kind guidance. The authors have added some conclusions about the correlations of the variables to the experiment results.

3.1 Effects of ultrasonic frequency on the CO2 ultimate absorption amount and AR

According to the results, it can be concluded that the CO2 absorption time of cement paste gradually shortens, while the CO2 AR and CO2 AA further increases when the ultrasonic frequency gradually decreases from 40kHz to 28kHz and 20kHz. This is because when the ultrasonic frequency increases, the ultrasonic intensity will increases accordingly. When the innceased ultrasonic intensity is excessive, there are excessive bubbles produced, which conversely increases attenuation of scattering, forming barriers of the ultrasonic wave. On the other hand, increasing of ultrasonic intensity will also lead to increasing of nonlinear attenuation, which is disadvantageous for uniform agitation. Consequently, particles of the cement paste are not well distributed. Therefore, when the ultrasonic frequency is 20kHz, the cavitation effect of ultrasonic agitation is best, and the corresponding CO2 AR and CO2 AA is highest.

3.2 Effects of ultrasonic vibration on the fluidity of cement paste after CO2

The above results indicates that fluidity of cement paste ultrasonically agitated is increased compared with that mechanically agitated. Cement particles in unit volume of cement paste is significantly increases when ultrasonic agitation is adopted, which means that ultrasonic agitation is better to break the flocculation structures formed by cement particles compared with mechanical agitation. Break of flocculation structures means that there are more free water in the paste to ensure fluidity of cement paste. However, both fluidities of cement paste agitated ultrasonically and mechanically decreases with an increase in CO2 AA and the decrement is significant. This is because the structure of the cement paste is gradually changes when CO2 AA gradually increases, which is disadvantageous for fluidity of cement paste.

3.3. Effects of ultrasonic vibration on pore distribution and porosity of hardened cement paste after CO2 absorption

3.3.1. The most available geometric diameter of pore distribution

It can be obtained according to Fig.6 that the most available geometric diameter of pore distribution of the hardened carbonized cement paste with ultrasonic agitation is lower than that with mechanical agitation. When the CO2 AA is respectively 0.44%, 0.88%, 1.32%, 1.76% and 2.20%, the most available geometric diameter of pore distribution of the hardened carbonized cement paste with ultrasonic agitation gradually decreases by 15.4%, 14%, 18.5% and 2.4% compared to that with mechanical agitation, which indicates that the most available geometric diameter of pore distribution of the hardened carbonized cement paste with ultrasonic agitation gradually decreases with an increases in CO2 AA. Therefore, the most available geometric diameter of pore distribution of the hardened carbonized cement paste can effectively be reduced with ultrasonic agitation.

3.3.2. The most available geometric diameter of pore distribution

It can be obtained from Fig.7, with an increase in CO2 AA, harmless pores and slightly harmful pores of the hardened carbonized cement paste will increases, while harmful pores and multi-harmful pores will decreases. However, increment of harmless pores and slightly harmful pores and decrement of harmful pores and multi-harmful pores is more significant when ultrasonic agitation is adopted.

3.3.3. Porosity

According to the above analyses, conclusions can be drawn that porosity of the hardened carbonized cement paste whether mechanically agitated or ultrasonically agitated decreases when CO2 AA increases. However, porosity decrement is more obvious when ultrasonic agitation is applied. Reasons for significant porosity decrement when ultrasonic agitation is applied lies in two aspects. On the one hand, cement particles can be minimized by ultrasonic waves, which is advantageous for hydration degree of cement particles. On the other hand, the flocculation structures formed by cement particles can be broken by ultrasonic agitation, which is advantageous for cement particles to get accesses to water to improve hydration degree. Improved hydration degree of cement particles indicating that more hydration products of C-S-H and CH are formed, which improves density of the hardened cement paste thus to reduce porosity of hardened cement paste.

3.3.4. Mean pore size

According to the above analyses, it can be obtained that both mechanical and ultrasonic agitation is helpful to minimize mean pore size and that mean pore size gradually decreases with an increase in CO2 AA. However, decrement in mean pore size when ultrasonic agitation is applied is more obvious. This is because the products of CaCO3 crystals are helpful to minimize pore sizes and optimize pore size distribution of the hardened carbonized cement paste.

3.4.2. Compressive strength

According to above experimental results, it can be obtained that increment in compressive strength of the hardened carbonized cement paste is more significant when ultrasonic agitation is applied compared with that when mechanical agitation is applied. Under ultrasonic agitation, cement particles are broken to smaller ones and flocculation structures formed by cement particles are broken by ultrasonic waves, which can promote cement hydration to produce more C-S-H. Therefore, the compressive strength is significantly increased by applying ultrasonic.

Point 2:The section 5 is very interesting, but it could provide some information regarding the chemical composition from SEM analysis, including the components in the transition zones between the paste and the aggregate.

Response 2:Thank you for the thoughtful suggestion and it can really make the paper more logical and complete. The authors have added an EDS analysis after section 4.2.

4.3 Energy dispersive spectrum(EDS)analysis

In order to better understand content changes of the elements of the paste after CO2 absorption by adopting ultrasonic vibration technology, EDS analysis was conduct the needle-like production respectively in cement paste without CO2 absorption, and cement paste with 1.32%CO2 absorption by mechanical agitation and that with 1.32%CO2 absorption by ultrasonic agitation, as illustrated in Fig.14a, 14b and 14c. Contents of each element are shown in Table 3.

According to Table 3, the main elements of the measure points respectively in Fig.14a, 14b and 14c are C, O and Ca. Meanwhile, there are also some Si and Al element. It can be obtained from Table 3 that contents of C, Ca and Si element are increased while that of Al element is obviously reduced when CO2 is absorbed by cement paste under mechanical agitation. However, under ultrasonic agitation, the contents of C, Ca element are increased more obviously and that of Al element continues to decrease.

It can be concluded according to the above results that with absorption of CO2, content of C element of the cement paste is increased and there are CaCO3 crystals in the cement paste. Moreover, due to the “cavitation effect” of ultrasonic agitation, CO2 is effectively dispersed in cement paste, leading to more CaCO3 crystals being produced by adopting ultrasonic agitation. Therefore, cement paste is able to absorb more CO2, thus forming more CaCO3, when ultrasonic agitation is applied compared with that when mechanical agitation is applied.

Point 3:The conclusions section should be completed with a general conclusion of the paper.

Response 3:The authors have revised the conclusion section as suggested.

  1. Conclusions

The results of our previous investigation of CO2 absorption performance of fresh cement paste indicate that the CO2 absorption rate and the ultimate absorption amount of fresh cement paste can effectively be increased. However, the mixing process of concrete is relatively very short in practice. Therefore, rapid and significant CO2 absorption by concrete cannot be realized only by reasonably setting the mechanical mixing rate and water-cement ratio.

As a consequence, a new method of ultrasonic agitation is applied to facilitate CO2 absorption performance of cement paste in this investigation. Meanwhile, the CO2 absorption rate and the ultimate absorption amount of cement paste under ultrasonic agitation is thoroughly studied. The results indicates that by applying ultrasonic agitation, CO2 absorption rate and the ultimate absorption amount of cement paste of cement paste can be effectively increased. Once the carbonized cement paste is hardened, the hardened cement paste is with increased compressive strength and decreased porosity and pore sizes, indicating refined pore structure of the hardened cement paste.

Due to limited investigation on CO2 absorption of cement paste, there are few literatures reporting about it. Moreover, due to limitation of the experimental device, only some critical problems are studied in this investigation regarding to the CO2 absorption of cement paste. However, there are still many problems need thorough investigations.

Reviewer 2 Report

In the reviewed paper authors presents the results of their investigations concerning the CO2 absorption by fresh cement paste using the ultrasonic vibration technology. Obtained results revealed that ultrasonic vibrating agitation not only can increase the CO2 but also can optimize the internal pore structure of materials and compressive strength of cement-based materials.

The paper requires the following amendments:

  1. The content of the introduction is overall and is poorly related to the topic of research carried out. The authors do not indicate the applied method's novelty compared to other methods used to increase the CO2 content in the fresh cement paste.
  2. There is no information about the applied ISO standard.
  3. In the experimental part, there is no information about the analytical methods and equipment used. How was the change in CaCO3 concentration during CO2 absorption studied? 

In my opinion, the reviewed manuscript may be accepted to publication in Appied Sciences after minor revision.

Author Response

Reviewer # 2:

In the reviewed paper authors presents the results of their investigations concerning the CO2 absorption by fresh cement paste using the ultrasonic vibration technology. Obtained results revealed that ultrasonic vibrating agitation not only can increase the CO2 but also can optimize the internal pore structure of materials and compressive strength of cement-based materials. The paper requires the following amendments:

Point 1: The content of the introduction is overall and is poorly related to the topic of research carried out. The authors do not indicate the applied method's novelty compared to other methods used to increase the CO2 content in the fresh cement paste.

Response 1:Thank you for pointing out the deficiencies. As far as the authors know, there is no related investigations on CO2 absorption of cement paste, which is proposed by our team. So far, there is only an authorized invention patent (a Mixing device for absorbing CO2 by using fresh concrete, China, CN201711123834.1) and a scientific report (Study on high-efficiency CO2 absorption by fresh cement paste. Construction Building Materials, https://doi.org/10.1016/j.conbuildmat.2020.121364) covering about the investigation on CO2 absorption of cement paste.

Therefore, the authors cannot compare the applied method with other methods used to increase the CO2 content in the fresh cement paste, because there are no other methods. The mixing process of concrete is relatively very short in practice, measurements such as setting proper mixing rate or controlling of water-to-cement ratio, or adding water reducer cannot realize rapid and significant CO2 absorption by cement paste.

At present, the authors have investigated influences of mechanical agitation and ultrasonic agitation on CO2 absorption of fresh cement paste. In the future investigation, the authors will further thoroughly investigate on the influences of other methods to affect CO2 absorption efficiency.

Point 2: There is no information about the applied ISO standard.

Response 2:Thanks for your careful review. The ISO standard mentioned in section 2.5.3 refers to the Chinese Standard of Method of testing cements-Determination of strength (ISO method).The ISO method is only a nomination, and it has no relationship with any ISO standard. The authors have quoted full name of the standard to avoid misunderstanding.

Details about this Chinese Standard are illustrated as follows:

GB/T17671-1999, Method of testing cements-Determination of strength (ISO method). Standards Press of China, Beijing, PR China, 1999.

Point 3: In the experimental part, there is no information about the analytical methods and equipment used. How was the change in CaCO3 concentration during CO2 absorption studied?

Response 3:① The authors have added information about the analytical method of fluidity determination, and other analytical methods are introduced in the experimental part.

The fluidity of Groups B and C was tested according to the Cement and Water Reducing Agent Compatibility Test Method (JC/T1083-2008) [25]. Pour the cement paste after CO2 absorption into the truncated cone and scrape flat. Then lift the truncated cone in the vertical direction and let the paste flow. 30 seconds later, measure the maximum diameters of the paste in two directions perpendicular to each other. The fluidity of the cement paste is average of the two maximum diameters.

â‘¡ The authors have added information about the equipment used.

The compressive strength and compressive strength at 3, 7 and 28 d were determined by a LS80-65–160 hydraulic compression tester to discuss the influences of ultrasonic vibration on the mechanical properties of fresh cement paste after CO2 absorption.

Pore structural distribution and porosity after 28 d were determined by mercury intrusion porosimeter a PoreMaster33 with a testing range of 3.5nm to 400μm.

Scanning electron microscopy (SEM) and energy spectrum analysis was performed with a scanning electron microscope Quanta 250.

â‘¢ As for the change in CaCO3 concentration during CO2 absorption, it is a valuable parameters regarding to the CO2 absorption performance of cement paste, which was not considered in the present investigation. The authors will consider investigating change in CaCO3 concentration during CO2 absorption in our future investigation to better understand CO2 absorption performance of cement paste.

Round 2

Reviewer 1 Report

In my opinion, the authors have modified all the suggestions presented by the reviewers so I think this document can be accepted for publication